# A 1-week diet break improves muscle endurance during an intermittent dieting regime in adult athletes: A pre-specified secondary analysis of the ICECAP trial

Jackson J. Peos[1]*, Eric R. Helms[2], Paul A. Fournier[1], James Krieger[3], Amanda Sainsbury[1]

1 Faculty of Science, School of Human Sciences (Exercise and Sports Science), The University of Western Australia, Crawley, Western Australia, Australia, 2 Sports Performance Institute New Zealand (SPRINZ), Auckland University of Technology, Auckland, New Zealand, 3 Weightology LLC, Issaquah, Washington, United States of America

* jackson.peos@research.uwa.edu.au

## Abstract

Athletes undergoing energy restriction for weight/fat reduction sometimes apply 'diet breaks' involving increased energy intake, but there is little empirical evidence of effects on outcomes. Twenty-six resistance-trained athletes (11/26 or 42% female) who had completed 12 weeks of intermittent energy restriction participated in this study. Participants had a mean (SD) age of 29.3 (6.4) years, a weight of 72.7 (15.9) kg, and a body fat percentage of 21.3 (7.5) %. During the 1-week diet break, energy intake was increased (by means of increased carbohydrate intake) to predicted weight maintenance requirements. While the 1-week diet break had no significant effect on fat mass, it led to small but significant increases in mean body weight (0.6 kg, $P<0.001$), fat-free mass (0.7 kg, $P<0.001$) and in resting energy expenditure, from a mean (and 95% confidence interval) of 7000 (6420 to 7580) kJ/day to 7200 (6620 to 7780) kJ/day ($P = 0.026$). Overall, muscle endurance in the legs (but not arms) improved after the diet break, including significant increases in the work completed by the quadriceps and hamstrings in a maximum-effort 25-repetition set, with values increasing from 2530 (2170 to 2890) J to 2660 (2310 to 3010) J ($P = 0.018$) and from 1280 (1130 to 1430) J to 1380 (1220 to 1540) J ($P = 0.018$) following the diet break, respectively. However, muscle strength did not change. Participants reported significantly lower sensations of hunger ($P = 0.017$), prospective consumption ($P = 0.020$) and irritability ($P = 0.041$) after the diet break, and significantly higher sensations of fullness ($P = 0.002$), satisfaction ($P = 0.002$), and alertness ($P = 0.003$). In summary, a 1-week diet break improved muscle endurance in the legs and increased mental alertness, and reduced appetite and irritability. With this considered, it may be wise for athletes to coordinate diet breaks with training sessions that require muscle endurance of the legs and/or mental focus, as well as in the latter parts of a weight loss phase when increases in appetite might threaten dietary adherence. Trial registration: Australian New Zealand Clinical Trials Registry Reference Number: ACTRN12618000638235 anzctr.org.au.

**Data Availability Statement:** All relevant data are within the paper and its Supporting information files.

**Funding:** This work was supported in part by Renaissance Periodisation and by the Australian Department of Education and Training via a Research Training Program Scholarship awarded to JJP, as well as by the National Health and Medical Research Council (NHMRC) of Australia via Senior Research Fellowships (1042555 and 1135897) to AS. The funders provided support in the form of salaries for authors AS and JP but did not have any additional role in the study design, data collection and analysis, decision to publish, or preparation of the manuscript. The specific roles of these authors are articulated in the 'author contributions' section.

**Competing interests:** AS reported owning 50% of the shares in Zuman International, which receives royalties for books she has written and payments for presentations at industry conferences; receiving presentation fees and travel reimbursements from Eli Lilly and Co, the Pharmacy Guild of Australia, Novo Nordisk, the Dietitians Association of Australia, Shoalhaven Family Medical Centres, the Pharmaceutical Society of Australia, and Metagenics; and serving on the Nestlè Health Science Optifast VLCD advisory board from 2016 to 2018. JK is employed by Weightology LLC. These affiliations do not alter our adherence to PLOS ONE policies on sharing data and materials.

## Introduction

The preparation for an athletic competition is one of the most challenging and stressful phases of an athlete's training year. This period often imposes high-volume, highly-fatiguing training regimes, whilst demanding unwavering discipline, focus and mental resolve. However, this period often coincides with a need to reduce body weight (notably fat mass) to reach a target weight class, or to improve the likelihood of contest success via aesthetic, biomechanical, or locomotive means [1,2]. To achieve the desired weight (and fat) loss, combinations of nutritional and exercise interventions are typically recommended, with the most common weight loss strategy implemented by athletes being continuous energy restriction [3,4]. It is well-documented that continuous energy restriction (often achieved via a reduction in carbohydrate intake) in athletes can result in decreased fat-free mass, reductions in muscle strength, endurance (presumably via observed decreases in glycogen stores) and reflexes, and increased appetite, irritability and fatigue, with such changes threatening adherence to the diet, making further weight and fat loss more difficult [1,2,5,6]. Thus, the effects of continuous energy restriction could jeopardise training and competitive success.

One strategy that could potentially solve the dilemma between the requirement to reduce weight and fat during preparation for competition, whilst also maintaining strenuous and high-volume training, is intermittent energy restriction. Although we recently demonstrated that 12 weeks of energy restriction, applied intermittently (3-week periods of moderate energy restriction alternating with 1-week 'diet breaks' involving increased carbohydrate intake to achieve energy balance) did not result in superior fat loss or retention of fat-free mass or resting energy expenditure compared to continuous energy restriction in adult athletes, we did show that it resulted in significantly lower sensations of hunger and desire to eat, and greater sensations of satisfaction (manuscript under review). Moreover, dropout from the intermittent diet group was approximately two-fold less than dropout from the continuous diet group (albeit this difference was not statistically significant, potentially due to the low number of dropouts overall), suggesting that the use of diet breaks to create intermittent energy restriction may facilitate adherence during weight loss among athletes. These findings come from measurements completed at the end of a period of energy restriction in athletes in both interventions—intermittent and continuous energy restriction (as opposed to at the end of the diet break). This timing of measurement was selected in order to standardise between the intermittent and continuous interventions. However, it is possible that appetite could be further reduced in the intermittent intervention during the 1-week diet break, reducing food-related distractions for improving mental focus on the competitive goal. This is an important consideration, as previous research has suggested that mental focus enhances sports performance while mental fatigue reduces it [7–10].

In addition to possibly enhancing mental focus, it is plausible that diet breaks might also positively impact training performance. Although we demonstrated that intermittent energy restriction did not result in sustained improvements in muscle performance (e.g., strength and endurance) compared to continuous energy restriction, it is possible that muscle performance could be transiently improved *during* the diet break. Indeed, as mentioned above, continuous energy (and carbohydrate) restriction is known to impair muscle performance (both strength and endurance) [11,12]. Thus, it is conceivable that a period of increased energy intake (by means of a carbohydrate-rich diet break) might offset these negative performance effects of energy (and carbohydrate) restriction, albeit temporarily. If so, this begs the question of whether diet breaks should be synchronised with key training sessions/weeks or mentally-demanding training blocks for an indirect competitive advantage, as we previously suggested [1]. This was proposed on the basis that the additional nutritional intake offered by the diet

break might temporarily suppress the adverse performance consequences of continuous restriction of energy (and carbohydrate) intake, resulting in a short-term improvement in performance [1]. Additionally, among athletes, anecdotal evidence suggests that diet breaks are a favourable time for high-volume and high-intensity training, with the short-term increase in energy (notably from carbohydrate) availability likely enhancing performance and reducing fatigue. However, this has not been tested empirically. Given the prevalent use of diet breaks among the athletic community as a perceived performance aid, rigorously testing the effects of diet breaks on performance is important. If diet breaks do *not* improve muscle performance as supposed, then their use could be unnecessarily delaying the attainment of the athlete's weight and fat loss goals, since we previously showed that intermittent energy restriction—with a 25% longer time requirement than continuous energy restriction—did not result in superior fat loss (albeit it was accompanied by reduced appetite, as mentioned above).

In light of the above considerations, the aim of this study was to test the immediate effect of a 1-week diet break (i.e., energy balance) after a period of energy restriction on a number of parameters we suspect may respond positively to the restoration of energy balance among resistance-trained athletes. Our focus is on changes from immediately before the diet break to immediately after the diet break in outcomes related to muscle performance, appetite and focus (e.g., muscle strength and endurance, sensations such as hunger and satisfaction, as well as irritability and alertness). In addition to these outcomes, we will also investigate fasting plasma concentrations of hormones that may regulate them.

## Materials and methods

A protocol design paper outlining the full study procedures for this trial and the weight-loss intervention, complete with inclusion and exclusion criteria, was previously published [13]. This trial was approved by the Human Research Ethics Committee at the University of Western Australia. Written informed consent was obtained from each eligible participant prior to inclusion. No data relating to individuals was identifiable in this trial.

### Eligibility criteria for participants

Eligible participants for this study were men and women who had completed 12 weeks of energy restriction, applied intermittently, during the ICECAP trial (manuscript under review), administered as 4 x 3-week blocks of moderate energy restriction interspersed with 3 x 1-week blocks of diet breaks involving energy balance. For the ICECAP trial underlying this study, eligible participants were aged $\geq$ 18 years, had completed $\geq$ 2 resistance exercise sessions per week for the previous 6 months or more, and were not currently on any weight loss program. In total, 30 participants (14/30 or 47% female) met the ICECAP trial eligibility criteria and began the intermittent energy restriction intervention after providing written informed consent, with 4 dropping out from the trial prior to completion of the 12 weeks of energy restriction in this intermittent arm. Accordingly, 26 participants (11/26 or 42% female) were available for the current study on the effects of the 1-week diet break.

### Dietary intervention

Participants underwent the 1-week diet break after having undergone 12 weeks of energy restriction, intermittently applied, in the ICECAP trial as previously detailed [13]. In brief, participants were exposed to intermittent moderate energy restriction intended to cause approximate weekly weight losses (during energy restriction) of 0.7% of their body weight [13]. During diet breaks, participants were instructed to follow a diet that provided approximately 100% of weight maintenance energy requirements for 7 days (dietary composition detailed

below). Weight maintenance energy requirements were estimated for each participant based on age, sex, body size and physical activity, as described previously [13]. In addition to an individualised energy intake prescription, each participant was provided with targets for daily dietary protein, carbohydrate and fat intake throughout the whole trial (including the 1-week diet breaks), with meal frequency, meal timing and foods/drinks consumed to meet energy and macronutrient targets chosen by each participant according to their own preferences. Participants were instructed to consume 2.3 g of protein per kg of absolute body weight daily during all phases of the trial, including the diet breaks, while approximately 20% of energy intake was allocated to dietary fat, with the remaining energy intake being allocated to carbohydrate. During the diet breaks (including the final diet break under investigation in this study), intake of dietary fat did not change, meaning the increase in energy was totally derived from an increase in carbohydrate intake. The developmental process and rationale behind the dietary intervention used for the ICECAP trial, including the diet break under investigation in this study, have been published previously in our protocol for the trial [13].

## Study overview and outcome measures

In this study, outcomes were measured on two occasions one week apart: before the diet break and after the diet break. The 'before' measurement was made on the last day of the last of four 3-week energy restriction blocks during the 15-week intermittent energy restriction intervention of the ICECAP trial. The 'after' measurement was made the morning after the last day of a 1-week diet break. All of the outcome measures for this study were collected in the fasted state (after 10–14 hours of water-only fasting from 20:00 hours on the night before the morning of testing).

**Body weight, height, fat mass, fat-free mass, and resting energy expenditure.** As detailed in our published protocol [13], body weight was measured in the laboratory using a calibrated scale, while fat mass and fat-free mass were determined by whole-body dual-energy X-ray absorptiometry (GE Healthcare, Chicago, Illinois, USA). Height was measured using a stadiometer, consisting of a ruler and sliding horizontal headpiece. Resting energy expenditure was calculated by expired gas analysis using a metabolic cart system (Ametek, Berwyn, Pennsylvania, USA) from the average 1-minute value during the final 10 minutes of a 30-minute resting period.

**Appetite.** Current appetite sensations (in the fasting state) were measured via an online survey in a subset of participants only (n = 22 out of 26), because there were 4 participants for whom the survey responses were not submitted. This survey was designed in line with previously published guidelines on good practice in carrying out appetite research [14]. The survey consisted of 8 items with each item scored on a continuous scale from 0 to 100, with questions pertaining to current feelings of hunger, desire to eat, how much food they felt they could eat (prospective consumption), satisfaction, fullness, as well as irritability, alertness and nausea. Specifically, the questions were: How hungry do you feel now? How full do you feel now? How strong is your desire to eat now? How much food could you eat now? How nauseous do you feel now? How irritable do you feel now? How satisfied do you feel now? How bloated do you feel now? How alert do you feel now? Participants moved the cursor along a horizontal visual analogue scale, anchored at each end with the statements "Not at all" or "Extremely low" or "None at all" (at the minimum score of 0) and "Extremely" or "Extremely high" or "A large amount" (at the maximum score of 100), to a point on the horizontal line that reflected the intensity of their current state.

**Hormonal regulators of fat mass, fat-free mass, resting energy expenditure, and appetite.** Venous blood samples for subsequent hormonal analyses were taken from a subset of participants only (n = 13 out of 26), because phlebotomy credentials were not obtained until

after baseline measurements were collected from the 13th participant beginning the final diet break in the intermittent energy restriction intervention. As detailed in the methods section of our randomised controlled trial (manuscript under review), blood was collected into EDTA-containing tubes and plasma was analysed for leptin, insulin like growth factor-1, testosterone, free 3,3′,5-triiodothyronine, active ghrelin and total peptide YY. The rationale for the hormones selected in our analysis was described previously [13].

**Muscle performance.** Muscle performance was evaluated by supervised endurance and strength tests using isokinetic dynamometry (Biodex Medical Systems, Shirley, New York, USA) as per our published protocol [13]. Muscle flexion and extension endurance at the knee (which assesses endurance of the hamstrings and quadriceps muscles of the leg, respectively), and at the elbow (which assesses endurance of the bicep and triceps muscles of the arm, respectively) were assessed during a maximum-effort 25-repetition set (total work, and work during the last third of the maximum-effort 25-repetition set). Meanwhile, muscle strength at the knee (hamstrings and quadriceps) and elbow (biceps and triceps) were assessed using a maximum-effort 3-repetition set (peak torque and power).

### Statistical analyses

As all outcome measures were continuous variables, comparisons in outcome measures taken before and after the 1-week diet break were made using paired Student's t tests, with the assumption of normality of the data being verified using a Shapiro-Wilk test. In the case of violation of the assumption of normality of the data, a Wilcoxon signed-rank test was used instead of a paired Student's t test. All analyses were completed with the statistical software JASP (Version 0.14). Effects of the diet break on outcome variables were considered significant when $P < 0.05$. Data are reported as means (with the lower to upper limit of the 95% confidence interval) unless otherwise specified.

## Results

Participants for the ICECAP trial underpinning this study were recruited between August 2018 and July 2019. There were 61 participants who began the ICECAP trial, 30 of which were randomised to the arm which was used in the current study (the intermittent energy restriction intervention). As 4 participants did not complete the intermittent energy restriction intervention, 26 participants were available to begin the 1-week diet break that forms the focus of the current study. All (100%) of the 26 participants who began the 1-week diet break completed the diet break and complied with testing requirements. Of these 26 participants, 15 (58%) were male and 11 (42%) were female, with a mean (SD) age of 29.3 (6.4) years, a mean (SD) weight of 72.7 (15.9) kg, a mean (SD) height of 173.2 (9.5) cm, and a mean (SD) body fat percentage of 21.3 (7.5)% before the intervention. As detailed in our report on the primary outcomes of the ICECAP trial, during the 12 weeks of energy restriction prior to the 1-week diet break under investigation in the current study, with that energy restriction being administered intermittently over 15 weeks, participants lost approximately 4.2 kg of body weight and 3.6% body fat compared to baseline (before energy restriction). Self-reported energy intake during the 1-week diet break was significantly increased compared to before the diet break (i.e., when participants were in energy restriction), by approximately 1770 kJ per day (Table 1).

### Fat mass, body weight, fat-free mass, resting energy expenditure, and hormonal regulators thereof

As per our intention for the 1-week diet break, there was no significant change in fat mass (in absolute terms or relative to body weight), or fasting plasma concentrations of leptin (a

**Table 1. Outcomes measured before and after a 1-week diet break following 12 weeks of energy restriction, applied intermittently over 15 weeks.**

| Measurement | No. | Before the 1-week diet break<br>Mean (95% confidence interval) | After the 1-week diet break<br>Mean (95% confidence interval) | P value[a] |
|---|---|---|---|---|
| Self-reported energy intake, kJ/day | 26 | 6950 (6310 to 7590) | 8720 (7900 to 9540) | **<0.001** |
| **Fat mass, body weight, fat-free mass, resting exergy expenditure, and hormonal regulators thereof** | | | | |
| Fat mass, kg | 26 | 15.2 (12.5 to 17.9) | 15.1 (12.3 to 17.9) | 0.278 |
| Fat mass, % | 26 | 21.3 (18.4 to 24.2) | 21.2 (18.4 to 24.0) | 0.383 |
| Leptin, pg/ml | 13 | 1910 (430 to 3390) | 2100 (130 to 4070) | 0.894 |
| Body weight, kg | 26 | 72.7 (66.5 to 78.9) | 73.3 (67.0 to 79.6) | **<0.001** |
| Fat free mass, kg | 26 | 57.6 (52.7 to 62.5) | 58.3 (53.3 to 63.3) | **<0.001** |
| Resting energy expenditure, kJ/day | 26 | 7000 (6420 to 7580) | 7200 (6620 to 7780) | **0.026** |
| Insulin like growth factor-1, ng/ml | 13 | 183 (161 to 205) | 196 (172 to 221) | 0.106 |
| Testosterone, ng/ml | 13 | 2.13 (1.17 to 3.09) | 2.28 (1.21 to 3.35) | 0.906 |
| Free 3,3′,5-triiodothyronine, pmol/L | 13 | 3.79 (3.34 to 4.24) | 4.01 (3.55 to 4.47) | 0.120 |
| **Appetite sensations (out of 100) and hormonal regulators thereof** | | | | |
| Hunger | 22 | 34.8 (23.7 to 45.9) | 19.2 (14.0 to 24.4) | **0.017** |
| Prospective consumption | 22 | 54.3 (41.6 to 67.0) | 41.0 (31.5 to 50.5) | **0.020** |
| Desire to eat | 22 | 41.8 (29.8 to 53.8) | 31.0 (23.0 to 39.0) | 0.069 |
| Irritability | 22 | 21.6 (9.10 to 34.1) | 10.7 (3.68 to 17.5) | **0.041** |
| Active ghrelin, pg/ml | 13 | 141 (78.1 to 204) | 155 (96.0 to 214) | 0.647 |
| Fullness | 22 | 41.5 (30.4 to 52.6) | 56.1 (45.3 to 66.9) | **0.002** |
| Satisfaction | 22 | 53.5 (42.5 to 64.5) | 69.6 (63.3 to 75.9) | **0.002** |
| Alertness | 22 | 47.1 (34.9 to 59.3) | 66.6 (57.7 to 75.5) | **0.003** |
| Total peptide YY, pg/ml | 13 | 93.1 (70.6 to 116) | 91.5 (63.5 to 120) | 0.830 |
| Nausea | 22 | 4.60 (2.00 to 7.21) | 4.60 (0.45 to 8.75) | 0.590 |
| **Muscle endurance during maximum-effort 25-repetition set, J** | | | | |
| Total work | | | | |
| Hamstrings | 26 | 1280 (1130 to 1430) | 1380 (1220 to 1540) | **0.018** |
| Quadriceps | 26 | 2530 (2170 to 2890) | 2660 (2310 to 3010) | **0.018** |
| Biceps | 26 | 1020 (840 to 1200) | 1040 (850 to 1230) | 0.363 |
| Triceps | 26 | 880 (730 to 1030) | 930 (770 to 1090) | 0.107 |
| Work during last third | | | | |
| Hamstrings | 26 | 339 (298 to 381) | 367 (323 to 411) | **0.018** |
| Quadriceps | 26 | 683 (587 to 780) | 713 (604 to 822) | 0.058 |
| Biceps | 26 | 274 (235 to 313) | 282 (242 to 322) | 0.183 |
| Triceps | 26 | 270 (215 to 325) | 284 (228 to 341) | 0.080 |
| **Muscle strength during maximum-effort 3-repetition set, N m** | | | | |
| Peak torque | | | | |
| Hamstrings | 26 | 96.7 (83.3 to 110) | 98.9 (84.8 to 113) | 0.097 |
| Quadriceps | 26 | 235 (159 to 312) | 236 (159 to 313) | 0.554 |
| Biceps | 26 | 54.7 (46.3 to 62.9) | 55.7 (46.8 to 64.6) | 0.407 |
| Triceps | 26 | 48.1 (41.4 to 54.8) | 49.0 (41.8 to 56.2) | 0.722 |
| Power | | | | |
| Hamstrings | 26 | 66.8 (58.4 to 75.2) | 68.3 (59.0 to 77.6) | 0.554 |
| Quadriceps | 26 | 123 (106 to 140) | 127 (109 to 145) | 0.407 |
| Biceps | 26 | 39.6 (32.9 to 46.3) | 40.2 (32.9 to 47.5) | 0.407 |
| Triceps | 26 | 34.9 (29.1 to 40.7) | 35.7 (29.5 to 41.9) | 0.407 |

[a], Bolded P values denote statistically significant differences from the value before commencement of the 1-week diet break at the significance level of $P < 0.05$.

hormone primarily released from adipocytes and correlated with total fat mass [15]), suggesting that the diet break was administered successfully. These data are shown in Table 1, as well as in Fig 1A–1C We have shown these data in tabulated as well as in figure (graphical) format because unlike Table 1 alone, the graphical format displays the trends in outcome measures for individual participants during the diet break, as well as the inter-individual variation, both of which are important. Compared to before the diet break, body weight, fat-free mass and resting energy expenditure were all significantly increased after completion of the diet break (Table 1, Fig 1D–1F), but these changes were not reflected by significant changes in plasma concentrations of insulin like growth factor-1 or testosterone (regulators of fat-free mass [16]) or free 3,3′,5-triiodothyronine (a modulator of resting energy expenditure [17]) (Table 1, Fig 1G–1I).

## Appetite and hormonal regulators thereof

Participants experienced significant reductions in appetite as a result of the diet break, as indicated by significant decreases in hunger and prospective consumption (but not in the desire to eat), and significant increases in fullness and satisfaction (Table 1, Fig 2A–2C, 2F and 2G). In line with a reduced appetite, irritability was significantly decreased, while alertness was significantly increased after the diet break (Table 1, Fig 2D and 2H), suggesting that the diet break may have relieved appetite urges that were mentally disrupting to participants. The significantly lower appetite, however, was not reflected by a change in fasting plasma concentrations of ghrelin (a modulator of hunger [18]), or total peptide YY (a major regulator of satiety [19]) (Table 1, Fig 2E and 2I).

## Muscle performance

In general, 3 of 4 indicators of leg muscle endurance (as measured by total work and work during the last third of a maximum-effort 25-repetition set in the hamstrings and quadriceps) significantly improved during the diet break, with the only exception being work completed by the quadriceps during the last third of the set (Table 1, Fig 3A, 3B, 3E and 3F). In contrast to the improvements in leg muscle endurance, there were no significant differences in markers of leg muscle strength (as measured by peak torque and power during a maximum-effort 3-repetition set) when comparing before and after the diet break (Table 1, Fig 3C, 3D, 3G and 3H). There were also no significant differences for any of the markers of endurance (work during a maximum-effort 25-repetition set) or strength (peak torque and power) in the arms (biceps, triceps) (Table 1, Fig 4A–4H). Overall, the results suggest that a 1-week diet break is effective for improving muscle endurance in the legs but not the arms, and is not effective for improving muscle strength.

## Discussion

This study showed that in adult athletes undergoing an energy-restricted fat-loss regime, a 1-week diet break (i.e., increasing carbohydrate intake to achieve energy balance) improved muscle endurance in the legs (but not the arms), and was accompanied by reduced sensations of hunger, prospective consumption, and irritability, and higher sensations of fullness, satisfaction, and alertness. While the 1-week diet break had no significant effect on fat mass, it significantly increased fat-free mass and resting energy expenditure. However, based on our previous work demonstrating that 1-week diet breaks do not enhance retention of fat-free mass or resting energy expenditure over the course of a 15-week fat-loss intervention (compared to continuous energy restriction), the observed increases in fat-free mass and resting energy expenditure during the diet break may be due to temporary replenishment of muscle

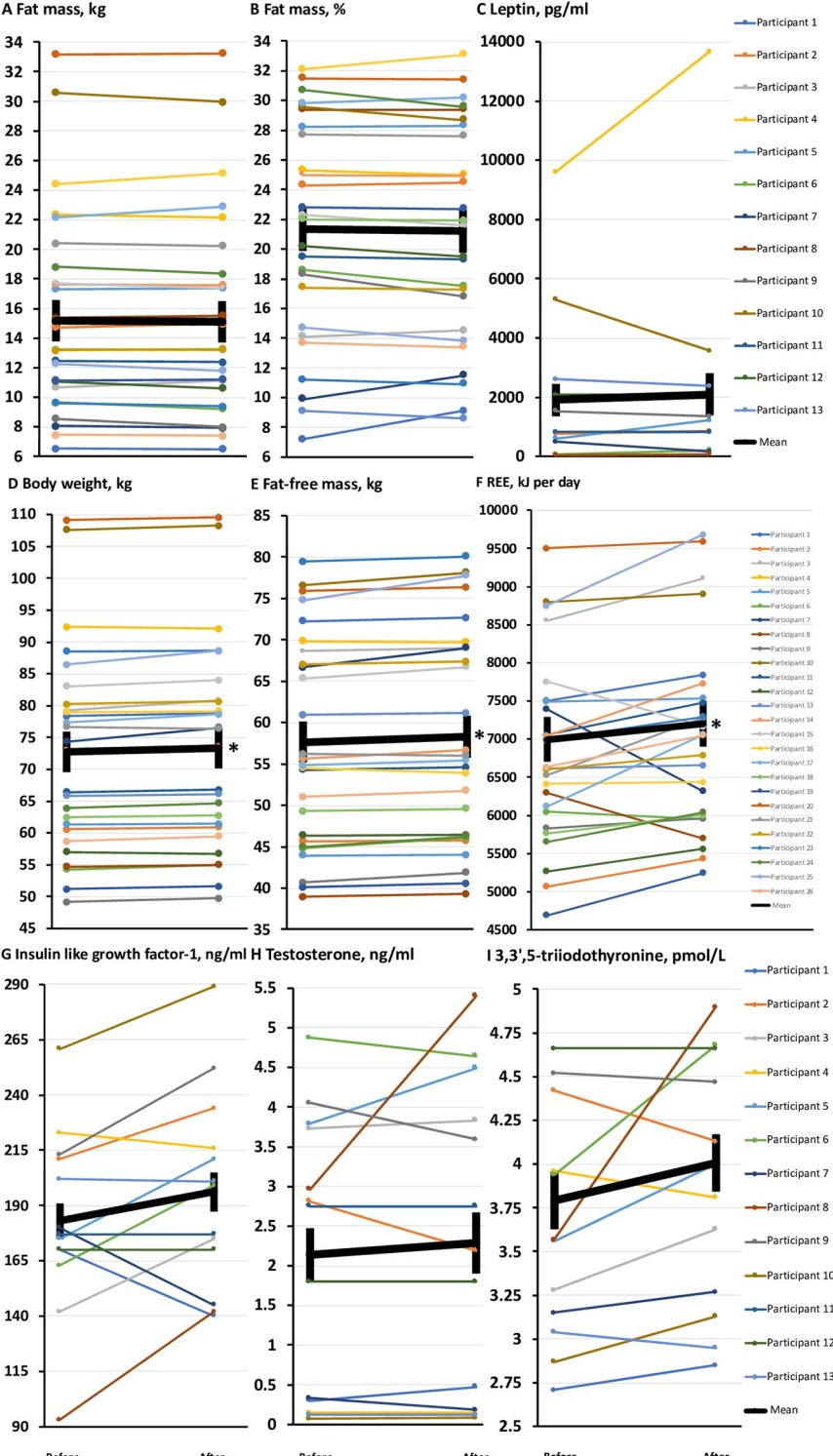

**Fig 1. Effects of a 1-week diet break on fat mass, body weight, fat-free mass, resting energy expenditure, and hormonal regulators thereof. (1A-1I)** Comparisons before and after a 1-week diet break in fat mass (kg and % of body weight), fasting plasma concentrations of leptin (pg/ml), body weight (kg), fat-free mass (kg), resting energy expenditure (REE, kJ/day), fasting plasma concentrations of insulin like growth factor-1 (ng/ml) and testosterone (ng/ml), and free 3,3′,5-triiodothyronine (pmol/L). Data are means ± SEM. * = significant difference compared to before the diet break, $P < 0.05$.

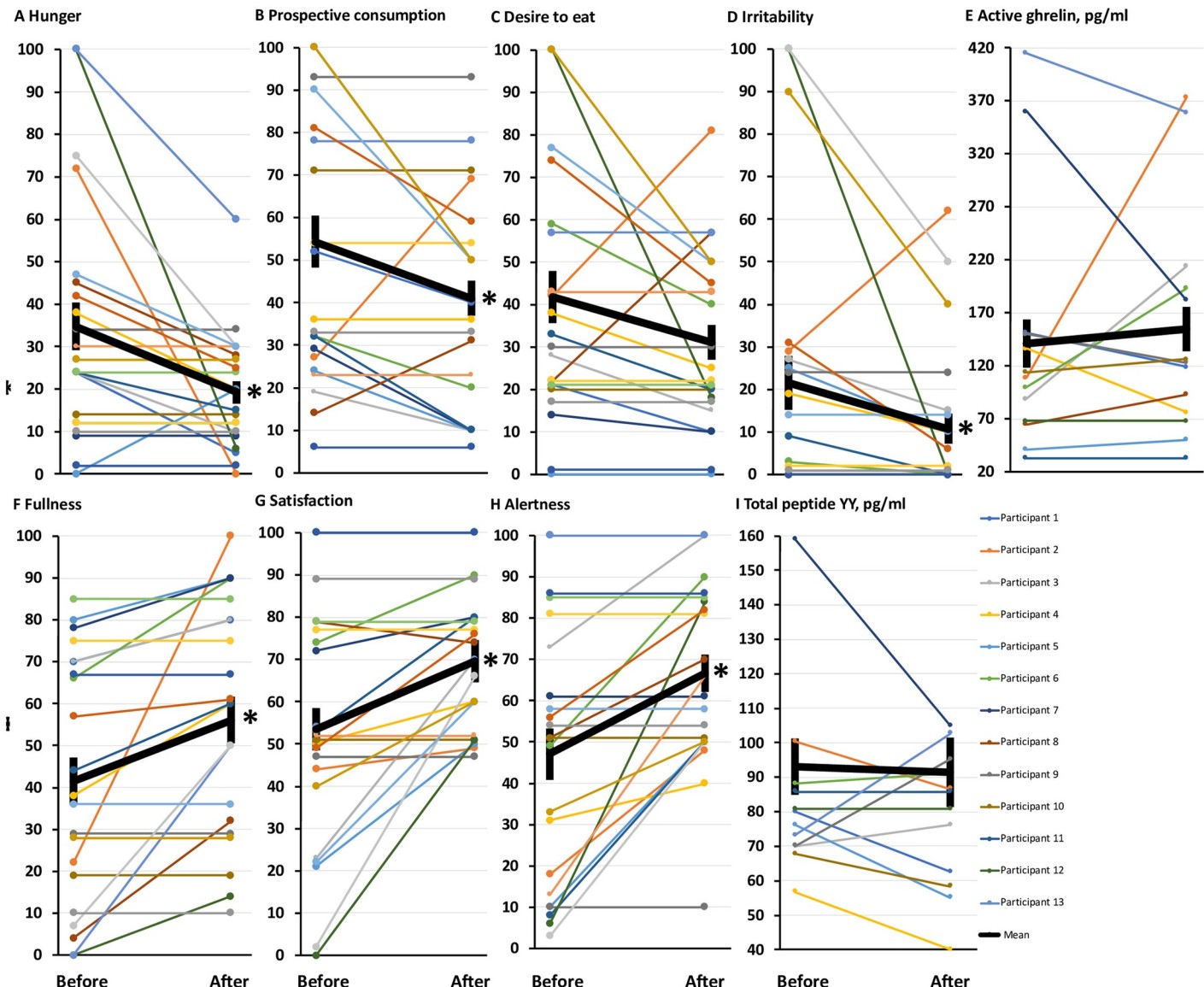

**Fig 2. Reduced sensations of hunger, prospective consumption and irritability, and increased sensations of fullness, satisfaction and alertness with a 1-week diet break. (2A-2I)** Comparisons before and after a 1-week diet break in sensations of hunger, prospective consumption, desire to eat, irritability, fullness, satisfaction and alertness measured by visual analogue scales in the fasting state, and fasting plasma concentrations of active ghrelin (pg/ml) and total peptide YY (pg/ml). Data are means ± SEM. * = significant difference compared to before the diet break, $P < 0.05$.

glycogen content [20–22], and an increase in dietary-induced energy expenditure resulting from the greater carbohydrate and energy intake during the diet break [23]. Overall, these results suggest that diet breaks may present an optimal time for maximising training intensity and volume (where the legs are involved), due to improved muscular endurance, as well as being an opportunity to engage in activities demanding alertness.

The findings of improved leg muscle endurance in response to the 1-week diet break support our previously-published suggestion that intermittent energy restriction might yield particular application to athletes by allowing the coordination of periods of energy balance with key training sessions [1]. Furthermore, given that mean body weight increased by only 0.6 kg during the diet break, we propose that it may be wise for weight-reduced athletes to finish

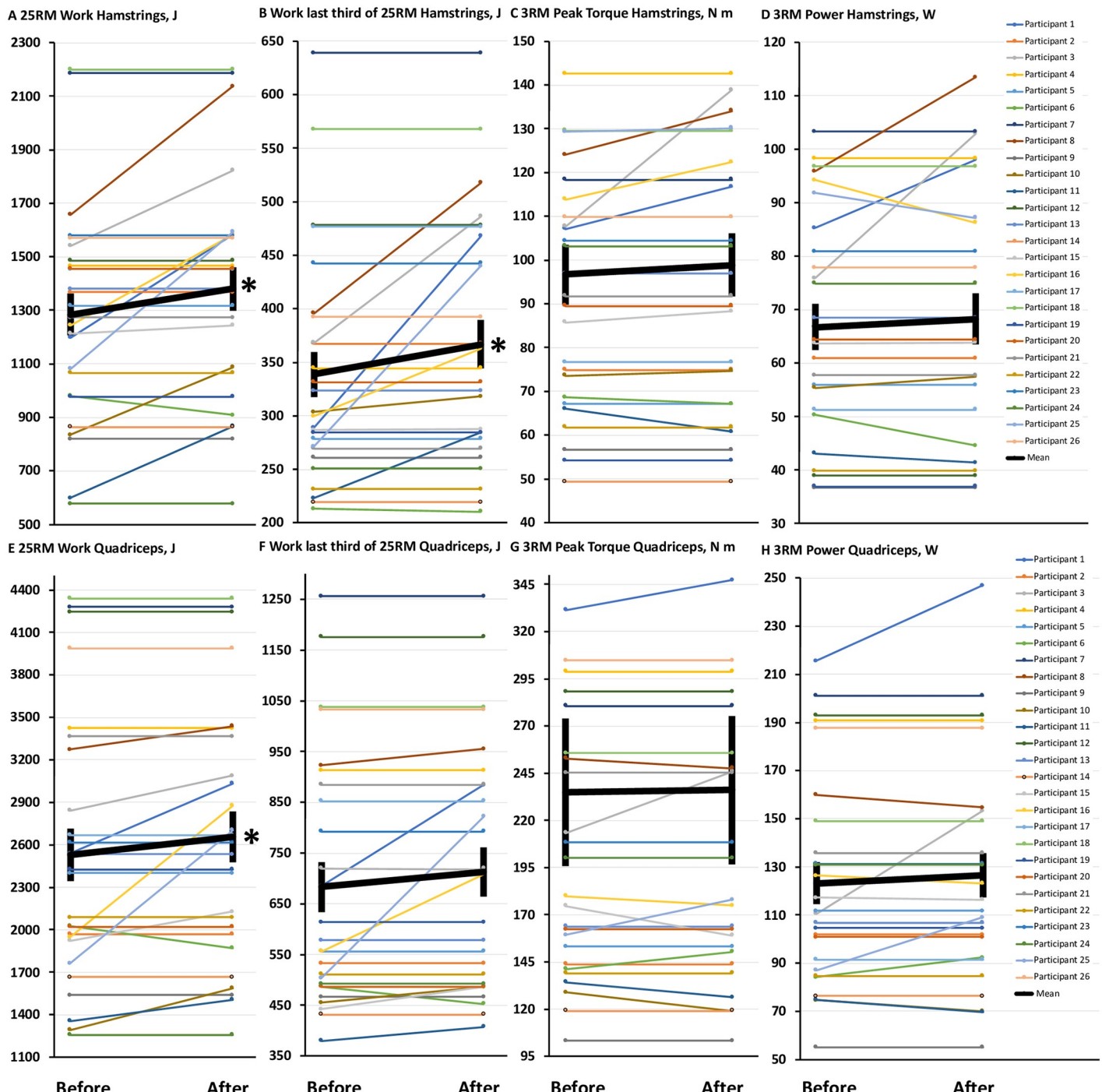

**Fig 3. Improved muscle endurance but not strength in the legs with a 1-week diet break. (3A-3H)** Comparisons before and after a 1-week diet break in muscle flexion and extension endurance at the knee (hamstrings and quadriceps) assessed using a maximum-effort 25-repetition set (25RM, total work, and work during the last third of the maximum-effort 25-repetition set), and muscle flexion and extension strength at the knee assessed during a maximum-effort 3-repetition set (3RM, peak torque, and power). Data are means ± SEM. * = significant difference compared to before the diet break, $P < 0.05$.

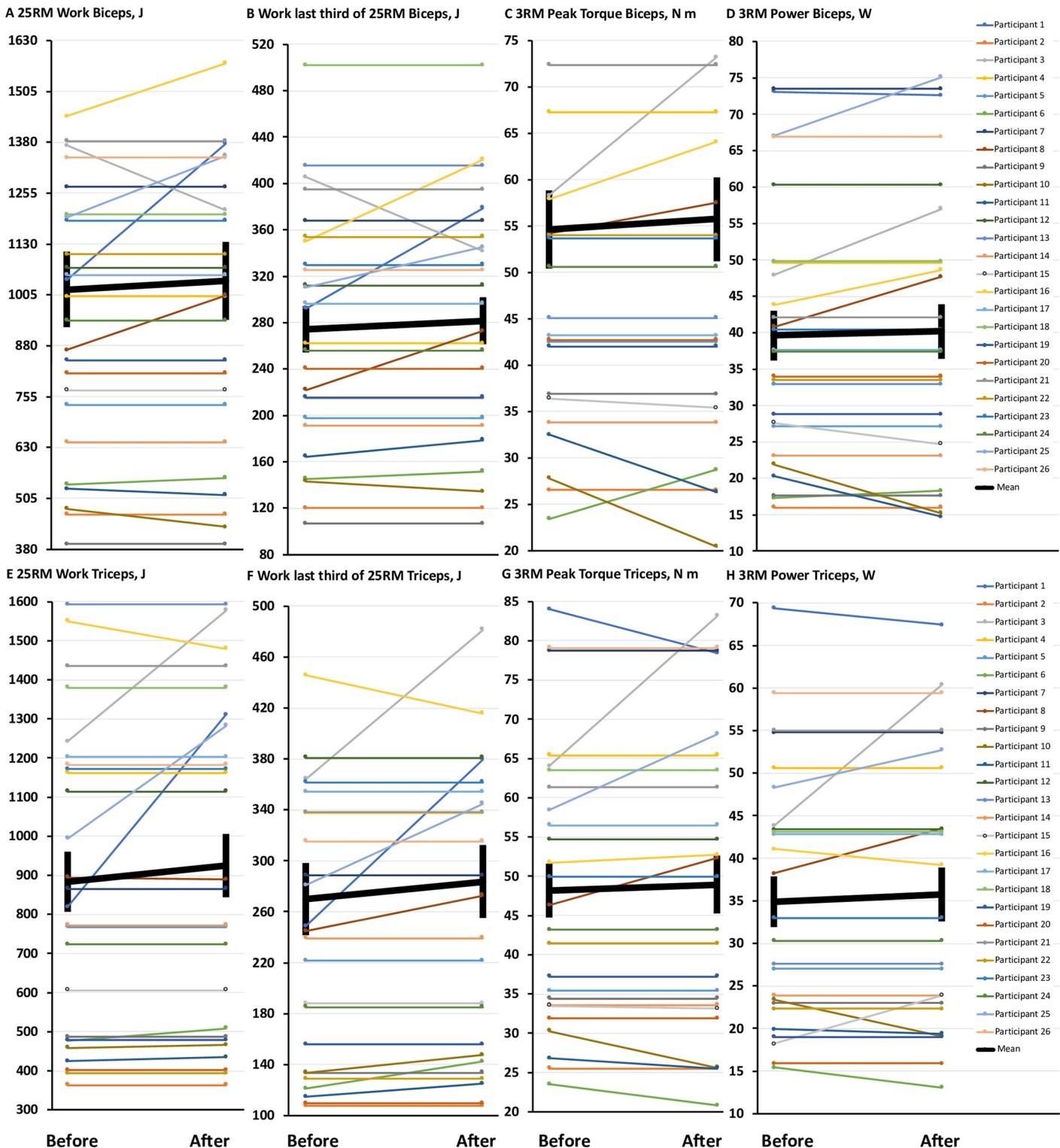

**Fig 4. No change in muscle strength or endurance in the arms with a 1-week diet break. (4A-4H)** Comparisons before and after a 1-week diet break in muscle flexion and extension endurance at the elbow (biceps and triceps) assessed using a maximum-effort 25-repetition set (25RM, total work, and work during the last third of maximum-effort 25-repetition set), and muscle flexion and extension strength at the elbow assessed during a maximum-effort 3-repetition set (3RM, peak torque, and power). Data are means ± SEM. * = significant difference compared to before the diet break, $P < 0.05$.

their weight loss phase one week prior to an actual competition, even if requiring a weigh in, so that muscular endurance in the legs could be improved in time for competition by means of a diet break. We supposed that diet breaks might provide athletes with adequate energy and carbohydrate availability for their training sessions, thus potentially negating the undesirable performance consequences often accompanied by energy and carbohydrate restriction [1,11,24–26]. Moreover, previous literature has demonstrated that inadequate energy and carbohydrate intake can impair muscular performance [11,24–26]. Indeed, in a recent study among athletes who increased their carbohydrate intake from less than 6 g per kg of body weight per day to 6 to 8 g per kg of body weight per day for three days, there were notable improvements in muscle endurance, as evidenced by a greater number of repetitions completed during a 12-minute exercise test [27]. With this in mind, it is reasonable to assume that the short-term increase in energy and carbohydrate intake—in the form of a 1-week diet break —might have reversed some of the unwanted performance effects arising from energy and carbohydrate restriction, resulting in the muscle endurance improvements that we observed.

It is uncertain why the improvements in muscle endurance that we observed in the legs were not reflected in the arms. In previous investigations it was observed that the legs lost proportionally less lean mass than the arms during energy restriction [28], and consumed less muscle glycogen than the arms during prolonged exercise (a 32% reduction in muscle glycogen content from prolonged exercise in the legs, versus a 69% reduction in muscle glycogen content from prolonged exercise in the arms) [29]. Thus, it is possible that greater depletion of muscle glycogen in the arms occurred during energy restriction and exercise among the athletes in our study, necessitating greater carbohydrate replenishment than that achieved during the 1-week diet break to elicit improvements in muscle performance.

Given the improvements in leg muscle endurance, we were surprised that leg muscle strength was not also improved during the diet break. The discrepancy between effects of the 1-week diet break on endurance and strength may be due to reliance on different energy systems for these two aspects of muscle performance. It is generally accepted that with an exercise period of maximal effort lasting up to 5 to 6 seconds in duration, the phosphagen energy system dominates, in terms of energy production to support the regeneration of adenosine triphosphate (ATP) [30]. Furthermore, energy yield from the phosphagen system is known to decrease rapidly as phosphocreatine stores are reduced, within 10 seconds of exercise duration [31]. When exercise continues for longer than a few seconds, the energy required to regenerate ATP is increasingly derived from blood glucose and muscle glycogen stores [32]. Thus, as our strength assessments lasted for approximately 5 seconds, which contrasts with the endurance assessments which lasted for approximately 30 seconds, it is reasonable to conject that the strength assessments predominantly relied on the phosphagen energy system, and for this reason may have been minimally affected by levels of muscle glycogen. Conversely, the endurance tests likely relied on the glycolytic energy system via consumption of blood glucose and muscle glycogen. With this considered, a greater carbohydrate intake during the diet break may have increased the reserves of carbohydrate for use by the glycolytic energy system, subsequently enhancing endurance performance and with no effect on strength. However, given the short duration of the endurance exercise bout and without any measurement of muscle glycogen levels, this interpretation should be taken with caution.

To our knowledge, this paper was the first to show that a 1-week diet break in athletes undergoing an energy-restricted fat-loss regime resulted in lower sensations of hunger, prospective consumption and irritability, and significantly higher sensations of fullness, satisfaction and alertness. This leads us to believe that coordinating diet breaks with periods of the fat-loss phase that require mental focus (e.g., key training sessions) may indirectly offer a competitive advantage. The relationship between hunger and mood has been previously investigated

[33], with results suggesting that higher levels of hunger are associated with signs of stress and lethargic behaviour. This is in line with anecdotal reports from athletes undergoing energy restriction, who report that hunger is not just a physiological, but also a psychological stressor. Literature supports this. Among a cohort of 371 student athletes, 34% reported that their athletic performance was negatively affected by hunger [34]. Furthermore, other studies investigating individuals undertaking energy restriction suggested that inadequate energy intake combined with other stressors (including exercise) results in degraded cognitive performance [35] as well as lower perceived work performance, poorer mood, and greater distraction [36]. This information collectively supports our findings of a reduction in drive to eat once energy intake was increased to energy balance (i.e., not energy restriction) during the diet break, and a consequent decrease in irritability and increase in alertness. This is an important finding for athletes considering that an athlete's mood is positively associated with competition success [37]. Further, finely attuned mental alertness (and lack of mental fatigue) is essential for athletes to reach their full performance potential [38], with mental fatigue often resulting in changes to behaviour including disengagement and decreases in motivation and enthusiasm [39]. Thus, diet breaks may offer a host of indirect competitive advantages to athletes by lessening hunger urges and food distractions—and consequently—threats to mood and mental focus.

Strengths of this study include the examination of both sexes, and the high retention of participants to the end of the 1-week diet break and data collection (100%). This study also has some limitations, namely the collection of blood samples from a subset of participants only (n = 13), as phlebotomy credentials were not obtained in time for the complete sample of participants. It is possible that the discrepancies between the statistically significant changes in fat-free mass, resting energy expenditure and hunger sensations during the diet break, and the non-statistically significant differences for hormonal regulators of these outcomes, could have been resolved with the complete cohort size. Furthermore, considering the absence of an independent comparator group, we cannot confidently attribute causality of the intervention in the same way that can be done with a randomised controlled trial. Thus, it is important that these results be interpreted with caution. We recommend that any future studies on this topic employ an appropriately-powered sample size and a controlled design with an appropriate independent comparator group to more fully elucidate the physiological and psychological impacts of diet breaks.

In conclusion, diet breaks could be a valuable tool for athletes during energy restriction. Acutely improved leg muscle endurance during the diet break could provide athletes with a competitive edge by offering an opportune time for high-quality, high-volume and high-intensity training involving the legs, while temporarily avoiding the performance decrement associated with energy restriction. Secondly, with notable reductions in drive to eat as a result of the diet break, athletes may exhibit less food-related distractions, facilitating a less irritable mood state and greater mental focus on the competitive goal.

## Supporting information

**S1 File.**
(CSV)

**S2 File.**
(CSV)

**S3 File.**
(CSV)

**S4 File.**
(CSV)

**S5 File.**
(CSV)

**S6 File.**
(CSV)

**S7 File.**
(CSV)

## Author Contributions

**Conceptualization:** Jackson J. Peos, Eric R. Helms, Paul A. Fournier, Amanda Sainsbury.

**Data curation:** Jackson J. Peos.

**Formal analysis:** Jackson J. Peos, James Krieger, Amanda Sainsbury.

**Funding acquisition:** Jackson J. Peos, Paul A. Fournier, Amanda Sainsbury.

**Investigation:** Jackson J. Peos, Eric R. Helms, Paul A. Fournier, Amanda Sainsbury.

**Methodology:** Jackson J. Peos, Eric R. Helms, Paul A. Fournier, Amanda Sainsbury.

**Project administration:** Jackson J. Peos, Eric R. Helms, Paul A. Fournier, Amanda Sainsbury.

**Resources:** Jackson J. Peos.

**Software:** Jackson J. Peos.

**Supervision:** Paul A. Fournier, Amanda Sainsbury.

**Validation:** Jackson J. Peos.

**Visualization:** Jackson J. Peos.

**Writing – original draft:** Jackson J. Peos.

**Writing – review & editing:** Jackson J. Peos, Eric R. Helms, Paul A. Fournier, James Krieger, Amanda Sainsbury.

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
