## [Decision Letter · Decision Letter 0]

9 Nov 2020

PONE-D-20-26641

A 1-week diet break improves muscle endurance during an intermittent dieting regime in adult athletes

PLOS ONE

Dear Dr. Peos,

Thank you for submitting your manuscript to PLOS ONE. After careful consideration, we feel that it has merit but does not fully meet PLOS ONE’s publication criteria as it currently stands. Therefore, we invite you to submit a revised version of the manuscript that addresses the points raised during the review process.

We look forward to receiving your revised manuscript.

Kind regards,

Cindy Gray, Ph.D.

Academic Editor

PLOS ONE

Additional Editor Comments:

In general, the manuscript is coherent and well written with very few typos. However, there are concerns about the extent to which we can rely on these findings given the small sample size. The authors need to show that the study is powered to detect the changes (or lack of) described. If it is not, then it needs to be described as a pilot study, and the results discussed accordingly.

Abstract: line 26 to 28 , should age not be used to describe the participants rather than “before the 1-week diet break” ? Similarly, is height required to be reported here? Lines 41 to 44 , the summary is confusing.

Introduction: Please could you describe your pilot study more clearly from line 62. The aim of the study doesn't appear to fully fill the gap, which to my reading had been set out as understanding physiological, performance, cognitive and psychological changes during the diet break, rather than at the end of it.

Method: Could you give the gender split of the participants in this study from the outset? Results should not be provided in the method ( table one). Furthermore, in table one titles, you can just say 95% confidence interval , you don't need “lower limit to upper limit”. Line 177 to 178, I don't think you need to refer to the protocol here, or at least the way you do it seems strange.

Results: Line 230, age should be provided as part of the description of the participants rather than as part of the baseline measures . Again, it is not clear that height is important – furthermore, its measurements is not described in the method. Line 238 to 240 seems to be in the wrong place? Given the low number of participants in the study , it is quite useful to have the graphs of individual participant outcomes, but it might be helpful to consider which of these should be in the main part of the manuscript and which should go into an appendix (there is a lot of information here)? Also, it would be good to refer to why individual outcomes are useful in the main body of the text, otherwise, perhaps Table 1 is enough? In addition, given only 13 participants provided blood samples, is it worth reporting these data here (particularly given the apparent high variability in some of the measures)?

Discussion: It appears a bit contradictory to read in line 324 that the results may lack practical significance for athletes and then go on to discuss the practical significance of the results. I am not sure that you need to keep referring back to the introduction in the discussion. Given the small sample size(s) in this study, it is important not to overplay the results. For example, to what extent is the study powered to detect the various changes? Should the next step be to repeat the study using a larger sample and a controlled design?

Journal Requirements:

2. Please include additional information regarding the survey or questionnaire used in the study and ensure that you have provided sufficient details that others could replicate the analyses. For instance, please provide additional information regarding the development and validation of the questionnaire, and if the questionnaire  is not under a copyright more restrictive than CC-BY, please include a copy, in both the original language and English, as Supporting Information.

3. Thank you for including your ethics statement:  "This study has been approved by the Human Research Ethics Committee at the University of Western Australia (RA/4/20/4340).".   

Please provide additional details regarding participant consent. In the ethics statement in the Methods and online submission information, please ensure that you have specified (1) whether consent was informed and (2) what type you obtained (for instance, written or verbal, and if verbal, how it was documented and witnessed). If your study included minors, state whether you obtained consent from parents or guardians. If the need for consent was waived by the ethics committee, please include this information.

"AS reported owning 50% of the shares in Zuman International, which receives royalties for books she has written and payments for presentations at industry conferences; receiving presentation fees and travel reimbursements from Eli Lilly and Co, the Pharmacy Guild of Australia, Novo Nordisk, the Dietitians Association of Australia, Shoalhaven Family Medical Centres, the Pharmaceutical Society of Australia, and Metagenics; and serving on the Nestlé Health Science Optifast VLCD advisory board from 2016 to 2018."

We note that one or more of the authors are employed by a commercial company: Weightology LLC.

4.1. Please provide an amended Funding Statement declaring this commercial affiliation, as well as a statement regarding the Role of Funders in your study. If the funding organization did not play a role in the study design, data collection and analysis, decision to publish, or preparation of the manuscript and only provided financial support in the form of authors' salaries and/or research materials, please review your statements relating to the author contributions, and ensure you have specifically and accurately indicated the role(s) that these authors had in your study. You can update author roles in the Author Contributions section of the online submission form.

4.2. Please also provide an updated Competing Interests Statement declaring this commercial affiliation along with any other relevant declarations relating to employment, consultancy, patents, products in development, or marketed products, etc.  

Reviewers' comments:

Reviewer's Responses to Questions

**Comments to the Author**

1. Is the manuscript technically sound, and do the data support the conclusions?

Reviewer #1: Yes

2. Has the statistical analysis been performed appropriately and rigorously? 

Reviewer #1: Yes

3. Have the authors made all data underlying the findings in their manuscript fully available?

Reviewer #1: Yes

4. Is the manuscript presented in an intelligible fashion and written in standard English?

Reviewer #1: Yes

5. Review Comments to the Author

Reviewer #1: An observational study was conducted to determine the effects of a diet break on fat and fat-free mass, resting energy expenditure, muscle endurance, hunger sensations, and irritability. A total of 26 athletes were studied. No differences in fat mass were observed pre to post diet; however, a small increase in body weight was noted. Muscle endurance of the legs improved after the diet break; however, muscle strength did not differ. Significant differences were observed with respect to hunger sensations, prospective consumption, irritability and sensations of fullness.

Minor revisions:

1- Line 218: Indicate the specific type of Wilcoxon test. The paired test, analogous to the paired t-test, is the Wilcoxon signed-rank test.

2- Cite the statistical software used for the analysis.

5- State and justify the study’s target sample size with a pre-study statistical power calculation.

6. PLOS authors have the option to publish the peer review history of their article (what does this mean?). If published, this will include your full peer review and any attached files.

Reviewer #1: No

---

## [Author Response · Author response to Decision Letter 0]

15 Dec 2020

All responses to specific reviewer and editor comments are included in the "Response to Reviewers" document attached

---

## [Decision Letter · Decision Letter 1]

5 Feb 2021

A 1-week diet break improves muscle endurance during an intermittent dieting regime in adult athletes – A pre-specified secondary analysis of the ICECAP trial

PONE-D-20-26641R1

Dear Dr. Peos,

We’re pleased to inform you that your manuscript has been judged scientifically suitable for publication and will be formally accepted for publication once it meets all outstanding technical requirements.

Kind regards,

Chris Harnish, PhD

Academic Editor

PLOS ONE

Additional Editor Comments (optional):

Great work!

Reviewers' comments:

Reviewer's Responses to Questions

**Comments to the Author**

1. If the authors have adequately addressed your comments raised in a previous round of review and you feel that this manuscript is now acceptable for publication, you may indicate that here to bypass the “Comments to the Author” section, enter your conflict of interest statement in the “Confidential to Editor” section, and submit your "Accept" recommendation.

Reviewer #1: All comments have been addressed

Reviewer #2: All comments have been addressed

2. Is the manuscript technically sound, and do the data support the conclusions?

Reviewer #1: (No Response)

Reviewer #2: Yes

3. Has the statistical analysis been performed appropriately and rigorously? 

Reviewer #1: (No Response)

Reviewer #2: Yes

4. Have the authors made all data underlying the findings in their manuscript fully available?

Reviewer #1: (No Response)

Reviewer #2: Yes

5. Is the manuscript presented in an intelligible fashion and written in standard English?

Reviewer #1: (No Response)

Reviewer #2: Yes

6. Review Comments to the Author

Reviewer #1: (No Response)

Reviewer #2: The authors thoroughly addressed the comments from the first review cycle. In particular, the authors added appropriate notes of caution regarding the scope of this data and the conclusions that can be drawn from a small sample size. The writing was clear and correct and the tables were well formatted.

7. PLOS authors have the option to publish the peer review history of their article (what does this mean?). If published, this will include your full peer review and any attached files.

Reviewer #1: No

Reviewer #2: No

---

## [Editor Report · Acceptance letter]

10 Feb 2021

PONE-D-20-26641R1 

A 1-week diet break improves muscle endurance during an intermittent dieting regime in adult athletes – A pre-specified secondary analysis of the ICECAP trial 

Dear Dr. Peos:

I'm pleased to inform you that your manuscript has been deemed suitable for publication in PLOS ONE. Congratulations! Your manuscript is now with our production department. 

Kind regards, 

on behalf of

Dr. Chris Harnish 

Academic Editor

PLOS ONE